# Impact of Digital and Non-Digital Urban Participatory Approaches on Public Access Conditions: An Evaluation Framework

**Thibaud Chassin** [1,2,*], **Adeline Cherqui** [2], **Jens Ingensand** [2] **and Florent Joerin** [2]

1 Laboratory of Geographic Information Systems (LaSIG), EPFL, 1015 Lausanne, Switzerland
2 Institute of Territorial Engineering (InsIT), University of Applied Sciences and Arts Western Switzerland (HEIG-VD), 1401 Yverdon-Les-Bains, Switzerland; adeline.cherqui@heig-vd.ch (A.C.); jens.ingensand@heig-vd.ch (J.I.); florent.joerin@heig-vd.ch (F.J.)
* Correspondence: thibaud.chassin@epfl.ch

**Abstract:** The gradual institutionalization of public participation increasingly compels local authorities to partially share their power over the transformation of urban areas. The smooth running of a participatory session is based on selecting the appropriate type of interaction, or medium, which supports the local authorities to reach and interact with a targeted public. However, local authorities often appear unfamiliar with the organization of interactive sessions with the population. This article introduces an evaluation framework that focuses on the access conditions of participants to the sessions of interaction. This novel perspective aspires to assist the local authorities in their decision to adopt a participatory medium (or method of interaction). Seven dimensions are investigated to this aim, namely accessibility, availability, adequacy, affordability, acceptability, awareness, and attractiveness (the last dimension is introduced in this article). In light of two real case scenarios that occurred in Western Switzerland, the use of the access framework is investigated for two potential purposes: (1) supporting the choice of a medium for an interactive session according to the urban project's context and the targeted public; and (2) improving future participatory approaches by assessing the representativeness of participants attending a past session in comparison to the originally targeted public.

**Keywords:** public access conditions; targeted public; urban participatory approaches; e-planning; urban planning; conceptual framework

## 1. Introduction

Over the past thirty years, the necessity of public participation and its institutionalization has been a growing challenge for governments [1]. This challenge is currently being accelerated and transformed by the integration of digital technologies [2]. In the context of local urban planning, public participation, which fluctuates between negotiation and information [3,4], is often multi-dimensional, both in the structure of the involvement of citizens and in the medium adopted by the authorities to broaden its reach [5]. Apprehended to lose power over the decision-making process [1], but also unsuccessful in the organization of direct interaction with the local population [6,7], elected officials often find themselves endorsing approaches designed by consultancy firms. However, these private actors are often foreign to the territory and are not as familiar with its specificities as the local authorities are. These social and spatial characteristics are a significant issue in participatory approaches [8]. The local authorities, with their detailed knowledge of the problems of the area in question, should therefore be more qualified to identify the targeted stakeholders for a participatory approach.

The identification of the appropriate population, known to add value to an urban planning project, meets two potentially antagonistic objectives: the legitimization of the

project as supported by elected officials [9], and the contribution of citizen expertise, i.e., the "citizen knowledge" [10]. To reach this targeted population by the participation process, the urban project managers (and the local authorities) need to identify the aspects of the project, which will be subject to negotiation in addition to the individuals to invite. For instance, considering a construction project for a center devoted to disabled senior citizens, these future users should be the target public of the participatory approach. The approach needs to be structured to mobilize this specific population. To encourage the targeted stakeholders (the project's future users) to attend a participatory session, project managers must adopt mediums (pictures, maps, digital or physical mock-ups, questionnaires, transect walks, 3D platforms, etc.) that will place the approach within the right context.

This article addresses these mediums (i.e., methods of interaction), for structuring citizen participation, from the perspective of the population access conditions. This specific lens investigates one of the numerous aspects of urban participatory approaches in detail. Hence, the tool introduced in this study should be included, not in a self-sufficient evaluation, but in a holistic evaluation that takes into consideration additional dimensions such as the overall cost of the approach, the political situation, the risks, the cultural context. After defining the layout of public participation, this article highlights the fundamental issues of citizens' access to the participatory process according to the medium selected by the planners and/or decision-makers. This contextualization of the stakes and challenges in designing participatory approaches supports the elaboration of an evaluation framework regarding access conditions of the public, and the leverages which can be implemented to foster its involvement. An increased knowledge of the relationship between methods of interaction and public access could facilitate the negotiation between planners and decision-makers in the medium selection, ultimately leading to an improved involvement of the targeted stakeholders, and their equality towards participatory approaches. This framework is then discussed using the examples of two participatory approaches articulated around two urban planning problems in Western Switzerland. Finally, the article examines the implied synergy between digital and non-digital mediums, besides their purposes as a vector for facilitating the engagement of the targeted public.

## 2. Public Participation Definition and Citizens' Empowerment

In all public domains, any approach that looks for citizens' inclusion in decision-making implies a lengthy process [11]. However, the participatory process should not be regarded as an additional layer of complexity in policy-making, but as an essential step to co-design a political decision. This study focuses on public participatory approaches initiated by authorities in the context of urban planning problems. We illustrated the components that compose these processes in Figure 1. An interaction session happens at a fixed time at a set place (real or virtual), during which citizens and authorities exchange information through a medium. A single session is a part of a broader process that will generate deliverables (e.g., reports, indicators) for an urban planning problem. Each session has a specific scope that aims to address particular components of the overall problem. These separate steps aim at reaching a particular audience (i.e., the targeted public), that should be adequately represented by the participants of the session.

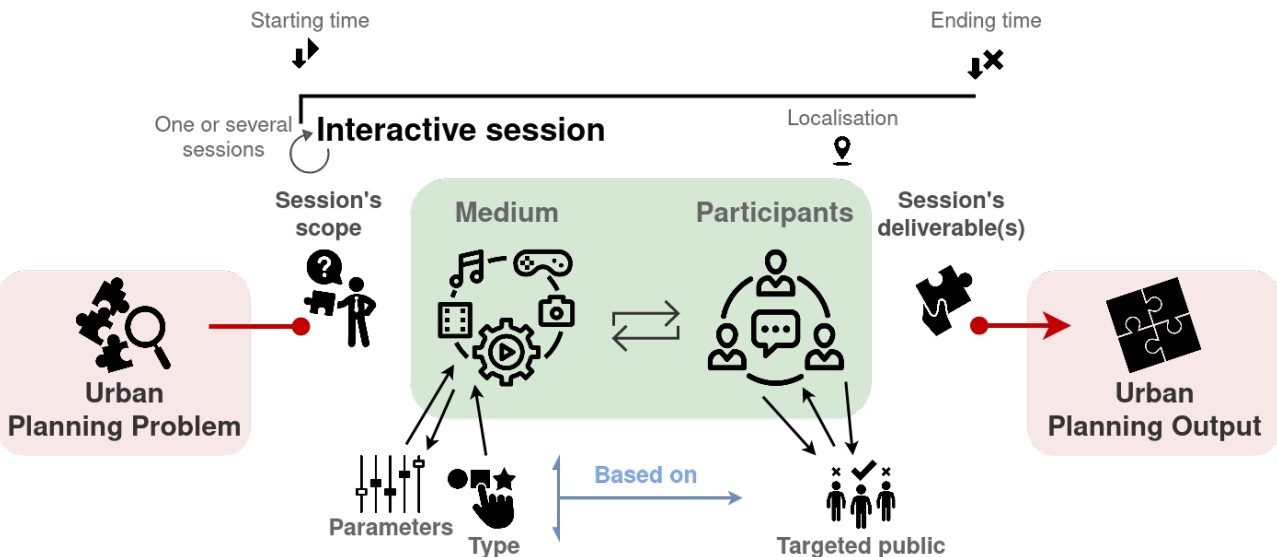

**Figure 1.** Schema of urban public participatory approaches initiated by authorities.

### 2.1. Public Involvement in Urban Decision-Making

Interactive sessions with citizens are increasingly adopted to address urban planning problems. This dynamic generates dialogues between the citizens who are involved as local stakeholders/experts because of their knowledge derived from experiencing their neighborhood [12], and the local authorities in charge of urban development. Nonetheless, local officials retain the legal power of decision-making.

Two concurrent phenomena characterize public participation in urban decision-making: (1) the stakeholders' need to participate in the decision, and (2) the compulsory practice of the stakeholders' involvement, enforced by national spatial laws, executed by local authorities [13]. Several examples demonstrate this second phenomenon in the western world that is wealthy enough to address these issues. In the United States, the establishment of the first Executive Order (EO), signed by former President Barack Obama in 2009, was endorsing the creation of the Open-Government Initiative, which contributed as a first step to reinforce the public's judgments in local decision-making [14]. However, this EO does not define the approach to gather the public's knowledge through citizen participation. This lack of detail is also observed in several Spatial Planning Acts (SPA) enforcing citizen participation in the planning processes. Such is the case in Quebec with the 2018 SPA (available on legisquebec.gouv.qc.ca, last accessed on the 17 May 2021) and its "dissemination of information, and consultation and active participation of citizens" (R.S.Q., c. A-19.1, art. 80.1); in France with the 2014 Lamy Act (available on legifrance.gouv.fr, last accessed on the 17 May 2021) where the territory needs to be "co-constructed" with inhabitants (LAMY, n°2014-173, art. 1 et 7); and in Switzerland with the 2019 SPA (available on admin.ch/opc, last accessed on the 17 May 2021) where citizens should be able to "participate adequately" in decision-making (LAT, RO 700, art. 4). These SPAs leave vast spaces for interpreting participatory processes, which leads to entrusting the local political actors with a decisive organizational choice over the design of the participatory approach. However, this empowerment can potentially be exploited to manipulate the population, and extend the power that the authorities have over the decision.

The gradual institutionalization of public participation also does not clarify the degree of delegation power, which has been measured in the literature through the different scales of citizen participation [3,4]. However, nowadays both politicians and literature acknowledge the necessity of reconsidering the modes of citizen involvement in projects that have a territorial impact, such as urban projects [15–17]. Such modifications of the peoples' living places [18,19] can lead to strong opposition if established without considering the project's social acceptability, based on stakeholder participation [15,20].

Regarding a public decision, if political-administrative actors decide or are obliged to share their decision-making power extensively, this does not automatically lead to the participation of a broad population. Several common obstacles include elitist control and alliances [21], lack of flexibility in urban procedures and non-integration of the deliverables from interactive sessions in decision-making [22], time limitation [23], acquiring the means to participate [24], seating limitation and ressources [25], participatory abilities [7], etc. All of these obstacles constrain the involvement of actors. New participatory mediums are thus being developed to partially address these barriers to promote democratized participation, i.e., reachable by everyone.

### 2.2. Digital Technologies to Support Participatory Approaches

In recent years, digital participatory mediums have become increasingly used. Since 2008, through the Civic Tech boom, a plethora of digital tools have aspired to revisit existing institutional processes [26,27]. These tools act as modern communication methods between authorities and citizens and deliver more direct, efficient, transparent, and fairer decision-making according to the founders of various Civic Tech companies [28]. This rise of Civic Tech coincided with the transition to a social web, namely Web 2.0 [29]. This web era allows users to consult web pages that create social interactions and networks between users, which implies the promotion of new means to participate. These tools are designed to facilitate the interactions between stakeholders and improve the feedback response time from the authorities, thus reducing the "participation fatigue" [30]. Examples of digital platforms enhancing the participation of the public have been evaluated and classified in the literature according to several factors, such as implemented functionalities, level of citizens' empowerment, and pricing strategies [31,32].

Among the technical pluralism of Civic Tech [33], Geographic Information System (GIS) tools appear valuable for urban participatory approaches. Indeed, the benefits of 2D GISs have been demonstrated for urban planning for many years [34,35], in particular, due to their ability to efficiently process and display geo-referenced content [36–38]. Mixing public-focused Web 2.0 components with GIS tools (e.g., Public Participation GIS) appears to be an opportunity for the authorities to gather an immense collection of citizens' feedback and to conduct broad surveys [39–41], i.e., to anchor citizens' local knowledge in its spatial dimension [42]. Early adopters have already acknowledged the usefulness of digital mediums in participatory approaches. According to the literature, users are eager to participate more often with collaborative online-mapping platforms with a user-friendly interface and a low entry cost (no software installation nor background knowledge required) [43]. New integrated methods including stakeholders throughout projects (from the design to the implementation) also emerge to produce local knowledge via participatory and collaborative mapping, notably in environmental management [44,45].

Besides these 2D digital mediums, Virtual Geographic Environments (VGE) and their third dimension have the potential to strongly impact the practice of urban participatory approaches. This additional dimension has many advantages, as described in the literature, including more accessible communication, intuitive representation, a better understanding of volumes, and increased immersion [46–48]. A better self-projection within a future urban project facilitates its overall understanding; therefore, VGEs appear to be an asset for achieving democratized citizen participation. A few guidelines exist in the literature about the representations and the functionalities that enhance urban participatory approaches [49]. However, the third dimension is employed only seldomly in urban projects and participatory approaches. A few examples are present in the industry CityPlanner, NeoCity, and the scientific literature [50–52].

Although Civic Tech leads to new opportunities in representation, interaction, and promotion of project elements with the introduction of new mediums, the significance of these stimuli (profitable or unprofitable) is not yet assessed regarding the democratization of urban participatory approaches. Numerous limitations prevail in the adoption of digital technologies, such as the blurring of the distinction between experts and laypeople, the

digital divide, the decision-making based on citizens' contribution, self-selection bias, laborious data processing, data security, junk information, and technical performance issues [22,30,48,50,53,54]. Despite the evidence that suggests that digital technologies represent one of the solutions that could bridge the gap between traditional tools and modern participatory approaches, providing a broad population with access to participatory processes does not seem entirely addressed by this digital medium.

## 3. Access Barriers in Participatory Approaches

In addition to outlining the foundations of urban participatory approaches, the previous section emphasizes the central position of the citizens/participants. In recent years, numerous governments have understood that building the public's trust in governance is achievable by being more transparent, responsive, and accountable. These values are notably advertised by the Open Government Partnership, and its declaration is now endorsed by 75 countries. Therefore, they are promoting the adoption of cooperation with citizens, notably in urban decision-making. Moreover, the involvement of assorted participants leads to the wisdom of the crowd, i.e., "the Many" surpass "the Few" with relevant and accurate contributions to any problem [55]. This concept emphasizes the significance of the group's heterogeneity and the individuals' independence in terms of values, experiences, and points of view. The participants' diversity in urban participatory approaches increases the number of different opinions, ideas, insights, choices, and aspects. A broader variety of participants could be reached via adopting various mediums, or providing distinct interaction modes.

From the authorities' perspective, this diversity means being able to effectively enroll 'the public', i.e., to reach the targeted public (also known as appropriate stakeholders) for the scope of an interactive session [56,57]. 'The public' differs in each context, and it englobes a full range of characteristics: elderly, youth, female, opponents, workers, residents, etc. The authorities seek to mobilize these targeted stakeholders to attend specific interactive sessions. This drive to action passes through the choice of a medium [58].

However, today the typical participant is frequently portrayed as a senior, male, educated and financially secure [59]. The over-representation of these socio-demographic characteristics in interactive sessions is not a choice artificially made by the authorities, but the result of a social positioning, i.e., a legitimacy to act. Consequently, the more individuals feel legitimate to act (interest in the issue, users affected by the project, involvement in public life, member of an associative network), the more they regard their voice as legitimate. One of the best-known illustrations of this phenomenon is the "Not In My BackYard" concept [60]. Furthermore, other social characteristics, such as job, income, capital, and speaking skills are also reinforcing this participation, which leads to the over-representation of a middle to upper social class in participatory approaches.

The scope of the session is highly versatile and requires appropriate stakeholders. Therefore, participants between different sessions should not be analogous. This gap between 'the public' and the participants attending the interactive session highlights a lack of representativeness, which is a critical parameter of public participation [61]. Low representativeness—linked to analogous socio-demographic characteristics or participants dissimilar to the targeted stakeholders—can lead to a bias (implied by the dominance of one group of stakeholders), the fatigue of the participants, and thereby to a decrease of the legitimacy of the decisions that have been made based on the results of the interactive session.

The choice of the medium used in an interactive session includes and excludes diverse parts of the population. Each medium has its benefits and its limits towards the citizens and the overall participatory approach in terms of location, duration, digital divide, communication, number of participants to manage, data, etc. These factors shape the collaboration between the authorities and 'the public' in terms of representativeness, which is conditioned by how each individual in the population is affected by the specificities of the selected medium. In this article, we argue that these hindrances can be identified by

evaluating the population's access to interactive sessions supported by the medium used in these sessions.

We introduce a framework that aims to assist the authorities to evaluate the population's access according to any type of participatory medium used in interactive sessions. We argue that the evaluation of public's access, if put into practice, leads to an improved representativeness. This framework can be useful at two stages: (1) before the participatory phase, to choose the medium according to the context and the targeted public (e.g., as a checklist of access issues to be considered), or (2) after the participatory phase, to assess the participants' representativeness in comparison to the public targeted by the authorities (continuous improvement process). The concept of access is well described in the medical literature from a healthcare/patient perspective. Levesque and al. identified more than ten articles addressing this topic through the definition of dimensions [62]. This inspiration from a medical background is relevant, considering the relation of healthcare vs. patients as state service vs. citizens.

In experimental medical sciences, the concept of access is defined using six dimensions: (1) *accessibility*, the distance between the location of the service and the customer; (2) *availability*, the ratio between volume (needs) and resources (capacity); (3) *adequacy*, the fit of the service to the client; (4) *affordability*, the service cost for the client; (5) *acceptability*, the customer opinion about the service; (6) *awareness*, described as the publicity of the service (use, role, goals, user target, etc.) and the information monitoring [63,64]. In the next section, we transfer these dimensions to an urban interactive session point of view.

## 4. A Framework for Evaluating Citizens' Access

This article aims to develop a framework that supports the evaluation, assessment, interpretation, and comparison of the access dimensions that are implied by the practice of a medium (Table 1). Access dimensions can, on the one hand, provide a useful tool for the authorities to weigh their choice to apply one medium rather than another to reach targeted stakeholders within urban planning problems. On the other hand, the framework can be used as an *a posteriori* review (with or without the participants) of the interactive session(s) that eventually take place, helping to improve the overall approach. In this framework, each access dimension is examined through a participatory approach lens.

**Table 1.** Scaled access dimensions present in the literature.

| Access Dimensions [ab] | Definition [ab] | Discrete Scale: Levels [1–5] [c] |
|---|---|---|
| Accessibility | Location |  |
| Availability | Supply and demand | |
| Adequacy | Organization | |
| Affordability | Financial and incidental costs | |
| Acceptability | Consumer perception | |
| Awareness | Communication and information | |

[a] Penchansky and Thomas 1981. [b] Saurman 2016. [c] Introduced in this paper.

### 4.1. Access Dimensions for Interactive Sessions

To characterize the dimensions, we outline each of them into five levels; the upper and lower boundaries of the levels will be labeled and illustrated in this section. Being context-specific, the definition of the dimensions within the framework is broad. Their estimation should be adapted to the situation by considering the targeted public as well as cultural, technological, and political factors. In addition to the framework's resilience over

specific contexts, the access dimensions assessment should also be considered through a multi-perspective approach, where age-segments should be evaluated, notably if the targeted public is varying. A younger and an elderly population would indeed have contrasted affinity to a particular type of medium (e.g., digital vs. non-digital).

Accessibility refers to the spatial effort that participants must undertake to reach the location of an interactive session. Four aspects determine this effort: (1) facilities, their presence or absence (parking spaces, transportation network); (2) distance (in kilometers, number of transits); (3) time, the duration of the journey (minutes, traffic jam, weather conditions); and (4) costs (financial, gas, parking, or transport tickets). The effort is weighted relatively low in the context of digital participation. The accessibility dimension has been defined from *unreachable* (level 1, the participant is not able to attend the session because the spatial effort is too significant) to *immediate* (level 5, the citizen can participate from any location via the use of a smartphone). An example of an average accessibility level in a typical Western European city would be a central location, such as downtown, which is easy to reach by public transportation and has convenient car parking.

Availability encompasses vacant seats for the public. These seats should not be regarded as an absolute number, but as the mobilization of the targeted public. According to the medium, this aspect is limited by physical attributes such as the room size, the number of facilitators, etc., or digital characteristics such as server uptime, traffic surges, latency, etc. This criterion has been delimited from being *closed* (level 1, no citizen can participate, for instance, an in-camera session with experts and elected representatives) to being *unrestricted* (level 5, the full targeted public can contribute without any restrictions, such as technical, social, financial or personal). An average availability should be considered as gathering the sufficient number of stakeholder representatives to mobilize a range of opinions or ideas from groups composing the targeted public (opposing and supporting).

Adequacy corresponds to matching the selected public to the interactive sessions, according to three concurrent aspects. The first aspect is matching the stakeholders' schedules to the timing of the interactive session. This aspect attempts to determine when a session takes place (part of the day), for how many hours, and with how many occurrences compared to the stakeholders' readiness. The second aspect is the fitness of the scope of the interactive session and its suitability to the citizens' interests and concerns. The third aspect is the participant's ability to participate (knowledge of participatory customs, willingness to speak in front of others, skills to operate a specific tool). The adequacy aspect ranges from *ignored* (level 1, an interactive session paired with its selected medium occurs without considering the public's idiosyncrasies) to *optimized* (level 5, the participation is shaped for the stakeholders). Average adequacy is designed to widen the scope of the involvement, with longer opening hours, shorter sessions, and acutely selected issues.

Affordability is the specific financial costs a citizen must incur to participate. The economic aspect is established using the cost of the session (occasionally used to reach a highly motivated population), combined with the organizational cost (i.e., childcare), specific material cost (i.e., virtual reality headset), or compensation (if applicable). The affordability factor ranges from *expensive* (level 1, an individual must pay a cost to participate, such as entry fee, materials, etc.) to being *rewarded* (level 5, the public is compensated for participating, for instance, financially, by goods, raffle, etc.). Average affordability would be when the financial cost to participate (transportation, material, fees, etc.) are balanced with the rewards (e.g., snacks and refreshments, reimbursed expenses, etc.); the participatory session is therefore economically even.

Acceptability characterizes the attitude that the targeted public has towards the participatory approach and, by extension, to the medium. This aspect is established on participants' social and cultural values. Its evaluation is, therefore, highly versatile. Several factors impact the acceptability, such as disadvantaged neighborhoods (regarding the interactive session's location), presence of specific individuals (other participants or local representatives), selected medium for the session, personal interest to participate (ideological engagement, political affiliation, geographical proximity to the project), and

the confidence towards authorities (expected results, the likelihood of participant's inputs being considered, transparency, etc.). The acceptability dimension ranges from *cautious* (level 1, citizens do not participate out of skepticism induced by concerns about the session or separation between the targeted public values and the session, including participatory washing, legitimacy, medium, and participants) to *trusted* (level 5, citizens eagerly participate because features of the medium (e.g., instant feedback) make it easier for the public to participate and comply with the session, inducing trust, respect, motivation, etc.).

Awareness varies according to the authorities' communication strategy. A public notice in a few lines on the municipal website compared to a massive presence on social media will not have the same impact on stakeholder groups. The communication strategy depends on the socio-demographic characteristics of the public targeted by the session. Awareness can be raised by resistance groups using their communication channels. Furthermore, the strategy strongly affects the acceptability dimension where information about the approach's philosophy can be broadcasted, such as the selected medium, the methods for analyzing the results, the scope of the interactive session, the session's rules, feedback frequency, the aspects subject to negotiation, and the integration of participants' inputs in the approach (and the urban project). The awareness factor ranges from being *ignorant* (level 1, no communication about the process is publicized to the broad public) to being *conscious* (level 5, a vast communication about the procedure is conducted and combined with an effort of mindful feedback and transparency, also including a co-construction approach of the process and the communication strategies with selected identified participants).

### 4.2. Attractiveness—A Meaningful Dimension of Access

The six previous dimensions of access are described from a citizen/medium relation. These dimensions are also defined in the literature from the perspective of participants' abilities [62]. Five abilities are categorized and bound to these dimensions of access: (1) the *ability to reach*, which involves self-mobility, is linked to the aspects of *accessibility* and *availability*; (2) the *ability to pay*, which regards the economic situation of the participant, is related to *affordability*; (3) the *ability to seek* is associated with *acceptability* and refers to personal freedom to select and adhere to the approach; (4) the *ability to perceive* is connected to *awareness*, and implies building intelligence and understanding regarding the interactive session; (5) the *ability to engage*, which fosters the opportunity to participate and the commitment to the approach, is coupled with *adequacy*. However, we argue that the dimension of *adequacy* does not entirely fulfill the *ability to engage*. Indeed, an adequacy that is optimized (in terms of timing, scope, and specific skills) does not ultimately result in the participation of the individual targeted by the authorities Concerns about user-friendliness and the appeal of the medium need to be taken into account. Therefore, we suggest translating these considerations into a seventh dimension of access that plays a vital role in the engagement of individuals, who are not only convinced by the project and the approach. This new dimension is the attractiveness of the interactive session, and, by extension, the attractiveness of the selected medium (Table 2).

Attractiveness is determined by dynamics of self-selection [9] and the willingness to participate [65]; both are strongly related to the motivation of the participants to commit themselves to contribute to the session. Their motivations, whether intrinsic (altruism, fulfillment, enjoyment) or extrinsic (community, ego enhancement, future return) [66], can be enhanced by an adequate implementation of the selected medium. Moreover, the perceived effort a targeted public has to invest to attend an interactive session also affects the attractiveness. This aspect can be translated as the entry cost of a session, i.e., the effort required from a stakeholder to participate, in terms of cognitive burden. This aspect is broken down into diverse factors: (1) the depth of exchanges (simple two-way interaction with a medium vs. full immersion with several individuals); and (2) the necessity to learn how to operate a specific tool (e.g., technology, methodology, rules, etc.). Several methods can be taken into account to increase the attractiveness, such as gamification elements (virtual or real), guerrilla participation (a term borrowed from UX design: guerrilla testing),

user-friendliness, and public acknowledgment. This dimension contributes to balancing the presence of individuals convinced by the approach (for or against an urban issue), who will always take part in the interactive session, by broadening the range of opinions. The attractiveness factor is classified from being *dull* (level 1) to being *appealing* (level 5), according to the targeted public.

**Table 2.** New scaled access dimension introduced in this paper.

| Access Dimensions | Definition | Discrete Scale: Levels [1–5] |
|---|---|---|
| Accessibility | Effort |  |

## 5. Contextualization of the Access Evaluation Framework

This section illustrates the use of the access framework in the context of two real case scenarios. The two examples presented hereafter depict, on the one hand, the evaluation of medium alternatives to select the most suitable medium for an interactive session, and on the other hand, the *a posteriori* evaluation of the participants' access following a specific interactive session.

### 5.1. Illustrative Case: Master Plan Revision of a Swiss Municipality, Anterior Access Analysis of Medium Alternatives

This project has been selected due to the direct involvement of the authors. The authors offered a proposal of collaboration to reply to a tender issued by the municipality. In that context, several discussions were conducted with the municipal officials resulting in the implementation of a proof of concept. The proposal of collaboration was ultimately unsuccessful. However, based on knowledge built during the exchange sessions and the prototype's development, this urban project has been selected to be used as an illustrative case.

#### 5.1.1. Project Description

Revising a city's Master Plan (MP) is a significant step in the development of an urban place. The MP has the objective to define areas to be built, conserved, or reconsidered during the future evolution of the city. This revision has two main challenges. First, all citizens are concerned, because the MP affects the entire space of the city; therefore, depending on the size of the municipality, the involvement of the population can be costly. Second, the elaboration of the MP is highly technical, which makes it arduous to include the average citizens. In this illustration, the Swiss city has a population of around 150,000 inhabitants. This Swiss city favored establishing a horizon 2030 'plan-guide' based on citizens' inputs to meet these challenges. The 'plan-guide' can be used as a valuable input to the MP revision, while being more flexible and less technical than the latter. This choice made by the municipal authorities is built upon nearly five years of experience in participatory approaches for urban issues. Indeed, this Swiss city started to regularly involve its citizens in urban problems following a massive project failure due to the population's non-engagement. The interactive session, as defined by the municipality, is intended to collect citizens' opinions about places that the public likes or dislikes within the city.

#### 5.1.2. Description of Medium Alternatives

In this framework for analyzing the access dimensions, three alternatives for collecting citizen feedback were discussed in depth during the meetings and partly planned by the authorities: (1) several transect walks; (2) an online survey; and (3) an online 3D platform. The 3D platform was the prototype put forth by the authors, and the online survey combined with the transect walks were the mediums ultimately adopted by the

municipality. These medium alternatives are presented in Figure 2 from the perspective of interactions between actors.

**Figure 2.** Schema from an actors' interactions perspective of the three options (Transect walk, 3D online platform, and online survey) considered for this urban problem.

Figure 2 portrays four roles engaged in these specific medium alternatives: *Participants*, *Facilitator*, *Authorities*, and *Transcriber*. The *Participants* represent the public affected or concerned by one urban project, and who attend the interactive session. The *Facilitator* often hired by the authorities or the session organizers, moderates and guides the participation. The *Authorities* are the local state entity that initiates the urban project; during a session, the authorities can be represented by a municipality deputy, a project manager, a decision-maker, or even an expert. Their responsibilities can include taking part in the design process of the session, and additionally answering citizens' concerns, appeasing the public by their presence, and sometimes assuming the role of the facilitator. The *Transcriber* (human or digital) symbolizes the data collection and the post-processing of all the information to report precise and neutral results of the session.

The mediums depicted in Figure 2 are described hereafter as conceptualized during the meetings between the local officials of the Swiss city and the authors:

Transect walks. Citizens interact simultaneously with each other, the authorities, and the facilitator (green arrows) during a walk within a neighborhood. Discussions are stimulated using elements encountered during the pathway. A mindful transcriber must gather information exchanged by actors to produce a report highlighting the main topics addressed. The density of data to be collected is challenging, as the participants do not monitor and edit their thoughts as in a written exercise, and several discussions may happen at the same time. Moreover, the reporting (of arguments, emotions, agreements, etc.) at the end of the session could misrepresent part of the stakeholders' debate (misunderstanding of the statements, omissions, emotional writing). The transect walks would occur on weekday evenings (starting time between 4 p.m. and 6 p.m., never the same day) for two hours. The starting and ending point locations would be different. Registration would be mandatory with the unlimited number of participants. Minimal information would be shared about the transect walks (distance, itinerary, topics).

Online 3D platform. All the stakeholders' interactions are centralized, i.e., citizens interact with other citizens, facilitators, and authorities through the platform. It could be accessed by the participants over a week, a month, or more. Synchronous and asynchronous

inputs by stakeholders would promote animated debates. Furthermore, the 3D scene that would be broadcasted, could evolve on-the-fly according to the public's insights (and authorities' readiness to make changes). In this medium, a digital script (i.e., the transcriber) would collect all data and produce automated indicators in real-time with the current session state, such as a dashboard of the participation status. Collecting raw data from the platform would prevent bias in the final report (e.g., the complete information can be gathered, no emotional influence, non-subjective interpretation, etc.) [67]. Along with this data, an overview report would be made available to the citizens. The prototype developed expressly for the interactive session aimed to assist stakeholders in locating points of interest in the city (the type of element, frequency of use, opinion). A comment section was made to promote free discussions around these locations. The creation of an account would not be mandatory to participate, yet strongly encouraged.

Online survey. The stakeholders' interactions are simplified. The survey is outlined by the authorities and completed by the participants. This medium enables only one-way interaction. Therefore, the complexity is lessened, i.e., the citizens must follow an URL link and answer the different questions. The survey could be online over several weeks, defined in advance by the authorities. Participants' replies would be administered remotely and automatically. A 3rd party survey service would be used. A dozen questions were sketched, including socio-demographic information collection.

### 5.1.3. Access Evaluation of Medium Alternatives

This hypothetical evaluation of medium alternatives illustrates how this framework should be applied *anterior* of the participatory approach to determine the most appropriate medium according to the context, but also to the segment of the population. The targeted population for this urban project was the entire population of the city. No restrictions were put on the appropriate stakeholders' socio-demographic characteristics. The authorities aimed to gather maximum information. Conducting participatory urban projects is not new in Switzerland. The population is thus used to participating or giving an opinion. Usually, an extensive communication campaign is conducted by the municipalities, and various channels are employed, such as social media, newspapers (digital and physical), a dedicated section on the city website, advertising panels, etc. The interactive session as designed by the authorities was intended to reach the entire population, all citizens having various socio-demographic traits, backgrounds, or situations. Therefore, assessing the access dimensions fluctuates, and levels are depicted within a range of minimal and maximal value (Figure 3). Indeed, the elderly's attitudes towards sessions planned on weekday evenings may contrast to those of people with jobs and family responsibilities. The argumentation of Figure 3 is based on Section 4 and detailed in Table 3.

The *ability* of the stakeholders *to reach* the interactive session is relatively high and met for the three mediums. The aspect of accessibility for the transect walks oscillates extensively according to the focused district. Each neighborhood has different amenities which increase or decrease the score of this dimension. In addition, a resident who wants to attend a transect walk in their district and another one downtown will be strongly favored. The survey and the 3D platform benefit from being online, i.e., reachable at any time from anywhere. They thus empower the participants with an effective *ability to reach*.

All districts of the city are considered in the MP revision, and the population is accustomed to participating. This context supports acceptability; however, each medium has its specificity. This aspect is deemed high for the transect walks due to their setup: the possibility to address concerns to the authorities without an intermediary. Regarding the online survey, anonymity is beneficial, allowing self-expression more openly without judgment from others. Nevertheless, the impersonal configuration of the survey demonstrates an aloofness that could be interpreted by the citizens as a disregard of the medium by the authorities. The 3D platform stands in the middle, where written elements saved and retrievable by anyone can be seen as an obstacle only by a part of the population (not accustomed to social media).

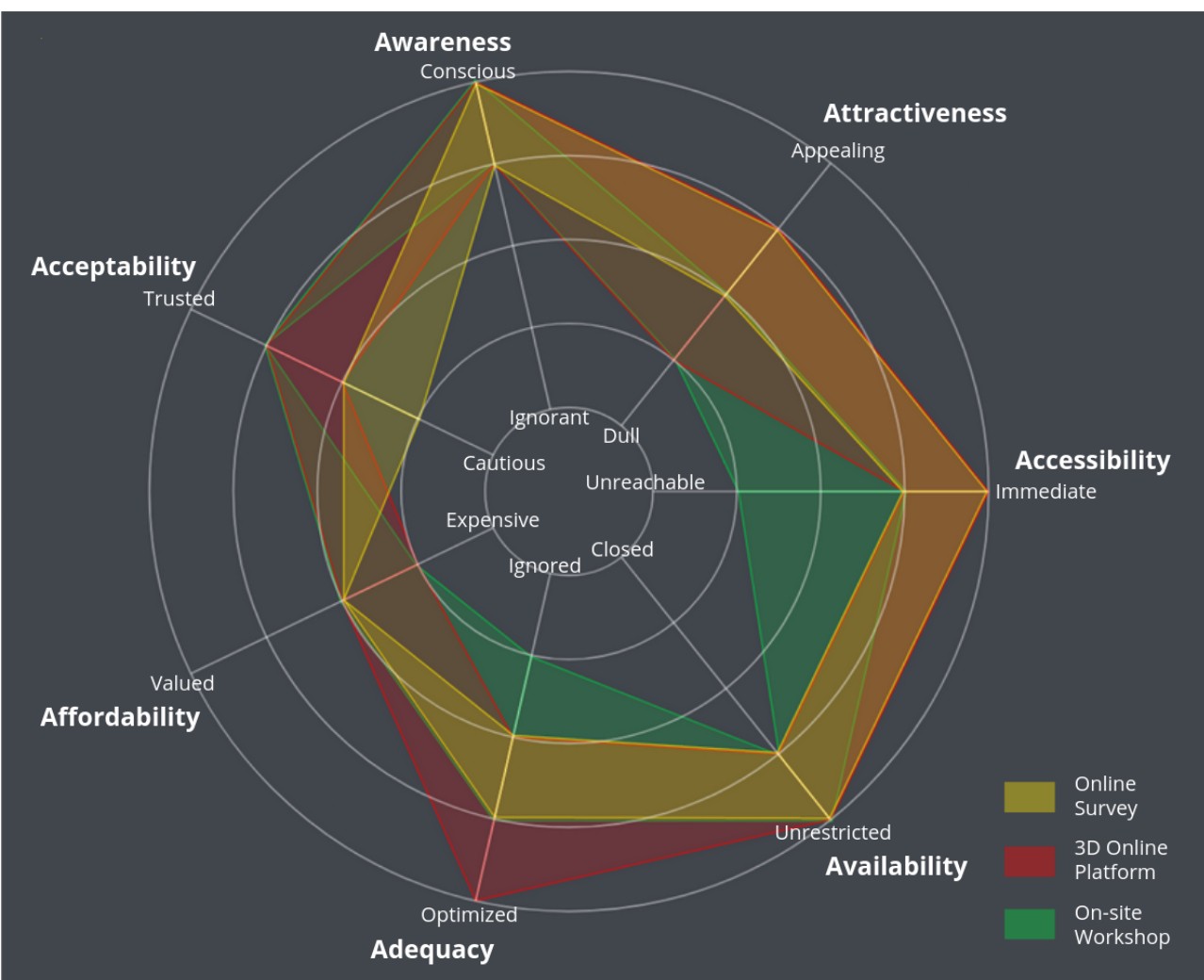

**Figure 3.** Dimensions estimation for the three alternatives studied for this urban planning problem.

Lastly, the *ability* of the population *to engage* differs according to the medium. The online 3D platform encompasses the most comprehensive range. As an online medium, citizens can participate when it suits them best. However, participants should be minimally proficient with 3D navigation, even if a small tutorial would be recommended before pinpointing locations. The attractivity dimension also differs depending on the participant. On the one hand, the 3D representations can be engaging by their novelty (especially for the youth), but on the other hand, repellent because of the non-familiarity of this type of portrayal (especially for the elderly). Transect walks are evaluated average-low according to adequacy and attractivity. First, several events would be organized, some starting at 4 p.m., which is inconvenient for daily workers. This medium setting, outside along a predefined pathway, is enjoyable, although quickly troublesome for people with reduced mobility (walk a couple of kilometers, no sitting for 2 h). The online survey achieves an average *ability to engage*. Despite being a medium well-known by the citizens, the questionnaire does not allow much flexibility in the thematics and the way of conducting the interactive session.

From this *a posteriori* evaluation of the dimensions of access, the authorities could decide to select a (or a combination of) medium(s) to enhance the mobilization of the targeted public. The medium that has the best overall score should be the preferred alternative, but according to other perspectives, such as costs or risks, another alternative can be selected. The criteria collected during the evaluation can be used to address any dimension that seems insufficient.

**Table 3.** Side-by-side comparison of medium alternatives facing access dimensions, argumentative table.

| Access Dimensions | | Evaluation Criteria (• Neutral, + Positive, − Negative) |
|---|---|---|
| **Accessibility** | *Transect Walks* | − need to travel to the location of the participatory session<br>− the starting point and ending location are different<br>• transportation facilities depend on the transect location |
| | *3D platform* | + from anywhere with a smartphone or from home with a computer<br>− not all smartphones can render 3D graphics (a computer is recommended) |
| | *Survey* | + from anywhere with a smartphone or from home with a computer |
| **Availability** | *Transect Walks* | + no limitation of participants (within an organizational limit) |
| | *3D platform* | + no limitation in the number of participants<br>− prototype, users may encounter a bug or downtime |
| | *Survey* | + no limitation in the number of participants<br>+ use of a 3rd party survey service, reduce the risk of downtime |
| **Adequacy** | *Transect Walks* | − thematics unknown in advance, chosen by the authorities<br>− 17 sessions of 2 h on weekdays in the evening (sometimes starting at 4 p.m.) |
| | *3D platform* | + platform accessible 24/7 via an internet connection<br>+ free choice to select the point of interest<br>• 3D navigation skills |
| | *Survey* | + platform accessible 24/7 via an internet connection<br>− non-flexible questions |
| **Affordability** | *Transect Walks* | • event on a weekday may induce organizational fees<br>• no entrance fee |
| | *3D platform* | − 3D graphics rendering requires a performant smartphone or computer<br>• no entrance fee |
| | *Survey* | • no entrance fee |
| **Acceptability** | *Transect Walks* | + citizens accustomed to the procedures and to participating<br>+ proximity of the authorities |
| | *3D platform* | + citizens accustomed to the procedures and to participating<br>• written record that can be consulted by all<br>• anonymity if not registered |
| | *Survey* | + citizens accustomed to the procedures and to participating<br>+ anonymity<br>− impersonal<br>− no opportunity to get more information on the consideration of the results |
| **Awareness** | *Transect Walks* | + massive communication campaign<br>+ various channels are exploited |
| | *3D platform* | + massive communication campaign<br>+ various channels are exploited |
| | *Survey* | + massive communication campaign<br>+ various channels are exploited |
| **Attractiveness** | *Transect Walks* | − outside, need to walk a couple of kilometers<br>− online registration is needed<br>− complexity of the actors' exchanges |
| | *3D platform* | + wow effect of the 3D representation<br>− non-familiarity with the medium<br>• one-way user interaction<br>• no mandatory registration but strongly encouraged |
| | *Survey* | + no registration<br>+ familiarity with the medium<br>− simple survey |

5.1.4. Outcome: Selection of the Medium

The municipality representatives eventually adopted a participatory approach combining two of the mediums presented before: online survey followed by transect walks. The

synergy between online and offline involvement promotes broader participation, where some citizens reluctant towards digital technology are willing to participate in a more traditional setup and vice versa. When we combine the evaluation of the online survey and the transect walks, we observe an expansion of some access dimensions, such as accessibility that now ranges from 2 to 5, or acceptability which scores 2–3 and 5. This expansion helps to counterbalance the limits of one medium with the benefits conveyed by the other. However, we can also notice that the overall lowest evaluation of the combination of these mediums leads to a decrease in the level of certain access dimensions, compared to the level of one or the other medium alone. Therefore, engaging the targeted public over the two mediums seems ambitious, but the prospect of committing individuals to one or the other medium appears reasonable. As explained throughout the description of the framework, the socio-demographic characteristics of each individual will affect their perception about the access dimensions. This variability should be mindfully considered during the evaluation of the dimensions in order to adequately mobilize the targeted public. If used effectively, this framework could assist the local officials to design the right interactive session by increasing the opportunity of collecting accurate information from the local experts, and limiting the risks of conducting a flawed participatory approach. Nevertheless, the practice of this framework is not a substitute for the monitoring of the participants' representativeness. If the socio-demographic groups attending the sessions are imbalanced, the local officials must respond by expanding the mediums, i.e., increasing the access, to engage the citizen groups that are missing.

### 5.2. Illustrative Case: ZImeysaver, a Posteriori Access Evaluation

This project has been selected due to its joint conduction between the Canton of Geneva and the institute InsIT of the HEIG-VD (institution of the authors). The documents used for this study are the deliverables produced as part of the participatory approach undertaken for this project [68].

#### 5.2.1. Project Description

The ZImeysaver is an industrial area that spreads over three municipalities: Vernier, Meyrin, and Satigny in the Canton of Geneva, Switzerland. The central part of the project is to create a new road section to connect a proximate highway with the ZImeysaver area. The goal is to improve the connectivity of the project site and to absorb the increased circulation caused by the future development of the industrial zone. The potential creation of 10,000 new jobs is projected for 2030. The concerned population is estimated at nearly 65,000 inhabitants (combined population of the three cities), plus day workers. The local stakeholders publicized a strong opposition towards the project. In this context, a participatory approach was set up to disarm the situation. The process was divided into three steps: a diagnosis, two interactive sessions (in the form of on-site workshops), and an informative session. Our *a posteriori* evaluation of the access will focus on the two interactive sessions.

#### 5.2.2. Interactive Sessions' Description

Two interactive sessions took place on 8 February and on 1 March 2018 from 6 p.m. to 9 p.m. The sessions were in line with a consultative level of participation [69] and adopted an on-site workshop as a medium of interaction. The red dot in Figure 4 highlights the location of the meetings. This position is adjacent to the ZImeysaver and is located within the most populated of the three affected municipalities. A train station connected to the two other cities is within walking distance (10 min). A bus stop is also close to the meeting point (5 min). Around 30 parking spaces are available outside the event building.

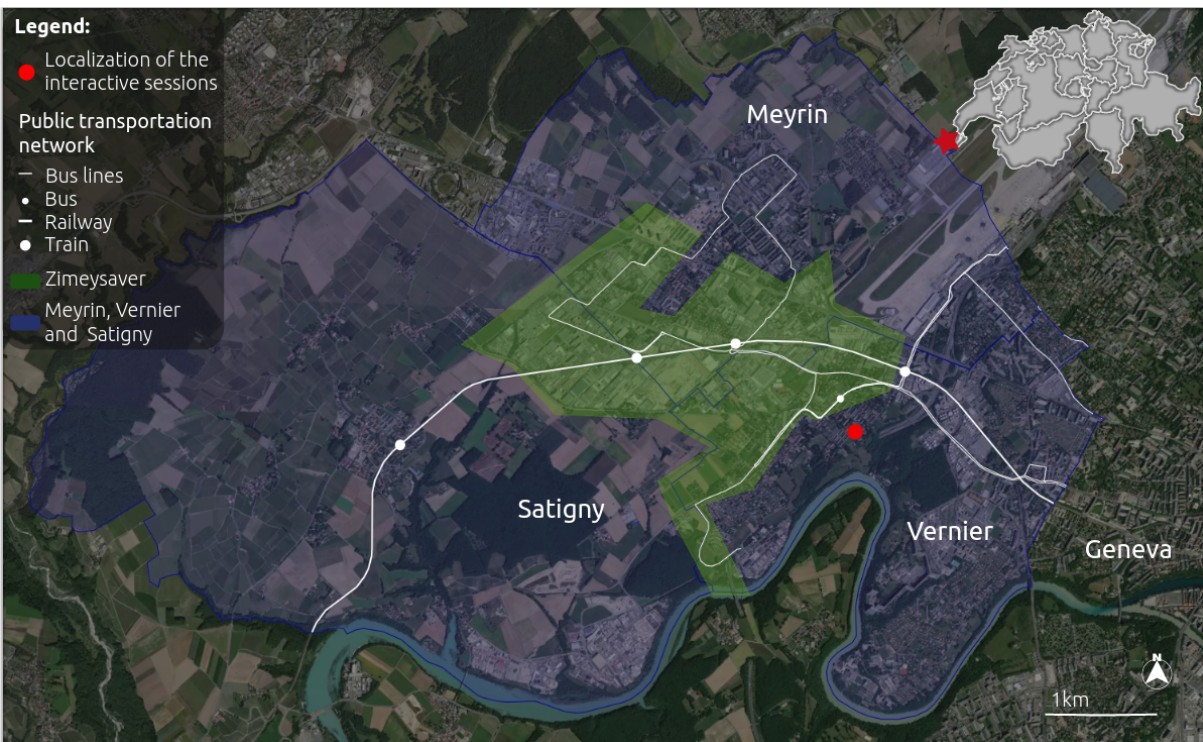

**Figure 4.** Contextualization of the interactive sessions designed for the ZImeysaver.

Each interactive session could host a maximum of 60 citizens (five round tables of 12 participants max). A round table was constituted of a dozen participants, a facilitator (project-managers, organizers), and a transcriber. The agenda was directed as described hereafter: welcoming of the participants, speech about the organization of the session, two times 40 min of exchanges about a specific issue (with a break between the discussions), immediate feedback done by the note-takers, closing speech, offering of snacks and refreshments. The issues discussed had been selected according to the citizens' concerns revealed during the diagnosis phase. The first interactive session focused on mobility and its impacts, while the second was axed on agriculture, landscape, and economy. To participate, citizens had to register for the session and select their preferred topics in advance. The communication strategy employed by the organizers was composed of a flyer distributed in all mailboxes of the citizens living in the municipalities and a post published on the Canton website (the Canton was scarcely active on social networks). The targeted public of the interactive session were ardent opponents; a few of them were already involved during the diagnosis phase. The two interactive sessions gathered 65 participants (46 for the first workshop and 19 for the second). These participants were mostly inhabitants and farmers affected by the project.

### 5.2.3. Access Evaluation

The *a posteriori* access estimation demonstrated herein highlights average-high access for these two interactive sessions (Figure 5). The goal of these sessions was to restore dialogue between the ardent opponents of the project and the authorities and to find a middle ground. Only 65 participants out of 120 (maximum capacity) attended the session; however, nearly all appropriate stakeholders came. The levels depicted in Figure 5 have been characterized according to the definition of each dimension described in Section 4, and their argumentation is demonstrated in Table 4.

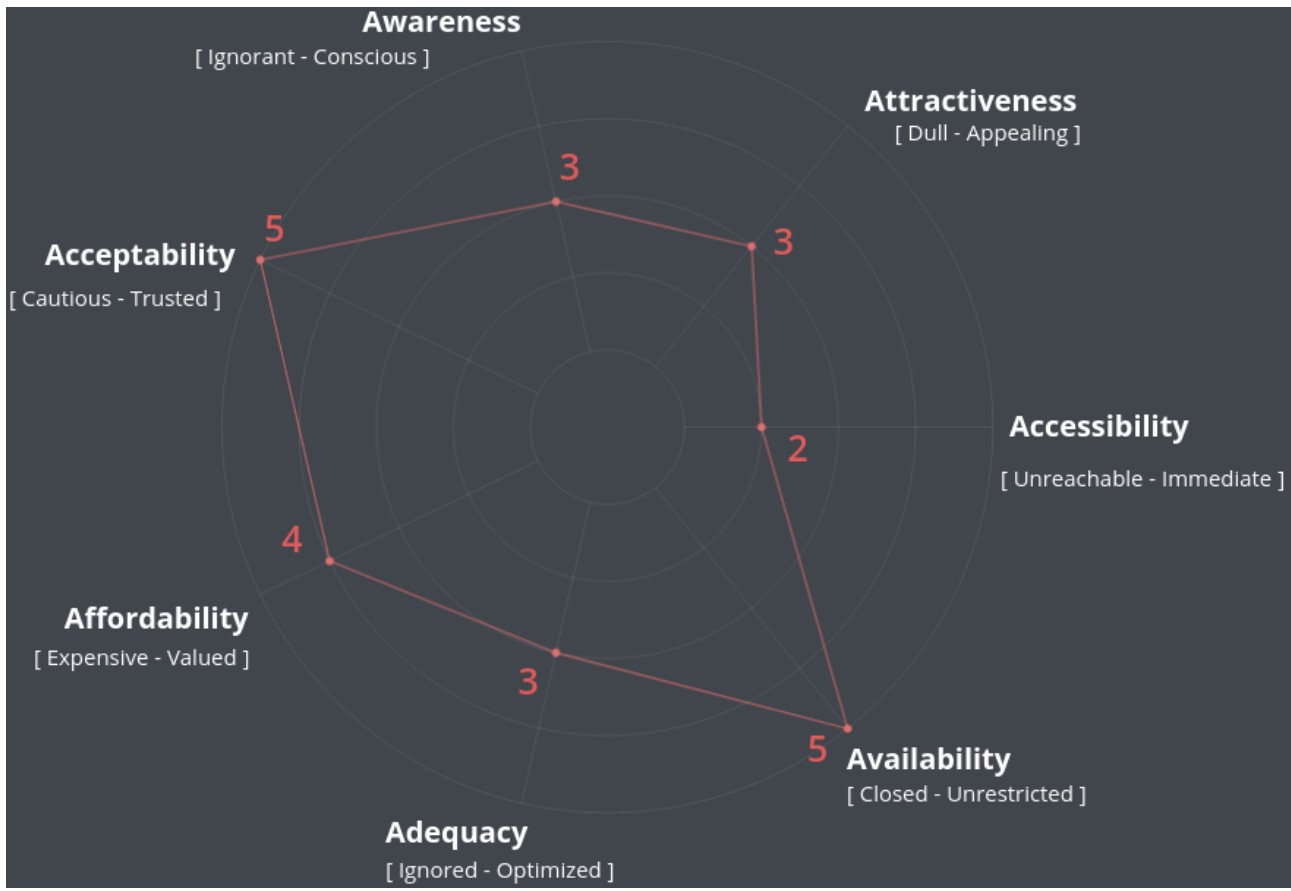

**Figure 5.** Dimensions estimation for the on-site workshop medium studied for the ZImeysaver project.

The lowest scaled dimension is accessibility. The second session indeed suffered from unfavorable weather conditions (snow), increasing the time of transportation. The organizers highlighted this argument as the main reason for the drop in attendance between the two sessions. A way to counterbalance this limited score would have been to increase the targeted public's *ability to reach* the interactive session by offering a new way to interact during the session remotely (videoconference) or after the session (mail post, email, online survey).

Regarding the average level of adequacy and attractiveness, the *ability to engage* the participants could have been enhanced. The choice of the on-site workshop medium limits the moment of engagement for a few hours on a specific day. Therefore, when a non-controllable factor drops the dimension of accessibility, the citizens can only sparsely participate. The public's profiles were individuals mostly over 50, some of them accustomed to the public arena (local politicians). The on-site medium with its formal settings fits their portraits (used to participatory codes, hypothetically more able to be free on evenings, highly motivated). However, the online mandatory additional step to register for the event adds a boundary to moderately motivated citizens.

The approach designed by the organizers fulfilled its role in mobilizing a handful of ardent opponents. However, the drop in attendance between the two sessions shows that some individuals (including probably a few opponents), who were interested in participating, could not take part in both sessions. Analyzing the criteria identified in the evaluation could help to understand this drop (in this case it was poor weather conditions) and to design potential solutions that could address future challenges.

**Table 4.** Argumentative table of the access dimensions' estimation.

| Access Dimensions | Evaluation Criteria (• Neutral, + Positive, − Negative) |
|---|---|
| **Accessibility** | − must travel to the location of the participatory session<br>− snowy weather for the event on 1 March<br>• well-connected to the public transportation network<br>• 30 parking spaces |
| **Availability** | + did not reach the maximum capacity<br>• 5 tables of a maximum of 12 participants |
| **Adequacy** | + topic selected according to citizens' concerns<br>− 2 sessions of 3 h on weekdays in the evening |
| **Affordability** | + snacks and refreshments offered<br>• no entrance fee<br>• event on a weekday may induce organizational fees (childcare, …) |
| **Acceptability** | + the authorities' open attitude: consultation<br>+ citizens accustomed to the procedures and to participating<br>+ proximity of the experts and project managers<br>• context of strong opposition towards the project |
| **Awareness** | + flyer distributed to all households concerned<br>+ some participants are already involved (word-of-mouth)<br>− no social network communication |
| **Attractiveness** | − online registration is needed<br>• formal setting of the on-site workshop<br>• full immersion increases the cognitive load |

### 5.2.4. Including Participants' Insights within the Access Evaluation Framework

The opinions of participants on the conditions of access to the interactive session can also be considered through this framework. Their assessment compared with the results of the previous section intend to create another perspective on the access dimensions (closer to the citizens). Therefore, expert and participants' views can be compared to highlight uniformities and dissimilarities. However, asking the participants to fulfill the raw access dimension framework would be too challenging. Therefore, the framework's complexity should be reduced. For this purpose, the authors argue for designing a survey where participants' socio-demographic characteristics and access dimensions (split into small queries) can be ascertained through specific questions. Each dimension could be symbolized by a thematic constituent of a few questions. The result would be a combination of the users' answers.

### 6. Access Evaluation Framework Implications for Participatory Approaches

The legal obligation to use participatory approaches in urban planning projects introduced by the SPAs encourages the authorities to reconsider their methods of governance. Facing this radical change, the local officials demonstrate hesitation based on concerns about their decision-making power (being the accountable entity with legal competency) or past negative experiences. This progressive institutionalization leaves the authorities to the design of these involvement processes without the expertise to conduct these types of approaches. The selection of a medium, as an instrument for social interaction and a transcriber of potentially antagonistic knowledge (between the technical knowledge guarantors on the one hand, and the local knowledge guarantors, that are experiencing the place on the other hand) is one of the critical steps to address for the design of these approaches. If mishandled, the mobilization of the targeted public to an interactive session can be missed, weakening the legitimacy of the solution provided by the entire participatory approach. The local authorities must carefully define the conditions of the access of the public to their participatory approach, in order to reduce, firstly, the power dynamics that occurs in

a group of participants, secondly, the over-representation of socio-demographic groups. Hence, the authorities should identify the targeted public and the means to adequately mobilize it through a medium. However, while these two steps strongly limit the power politics within participants by improving the heterogeneity of the group, they do not guarantee the absence of power dynamics [70]. The access evaluation framework developed in this study aspires to support the authorities in their choice of medium by evaluating dimensions around the conditions of access of the targeted public. Through the fulfilling of the seven dimensions, the local officials can estimate the medium that is most adapted to the targeted stakeholders of the session, including identifying benefits, opportunities, as well as disadvantages introduced by each studied alternative. Via this estimation, the authorities can efficiently arrange their expectations about the deliverables for the interactive session. Moreover, this framework applied as an evaluation tool following an interactive session can enhance authorities' understanding of the citizens' attitude towards a particular medium. This framework, therefore, aspires to help local officials to build confidence in their aptitude to design such interactive sessions by offering them the tools to succeed.

The harmonization of the territory's economic development with the image of this territory as a "living place" to be protected [16] is based on numerous parameters. These parameters determine the selection of a medium that fits the targeted public and the overall approach, such as: level of participation, level of interaction, purpose process of the stage, reach, scalability, participant selection, participant skills, cultural applicability, costs, interaction quality and depth, and required labor and expertise [71]. These factors, firstly described for online mediums, are also valid for on-site participation. This large set of factors ranging from the user's abilities to the project's specificities, and passing by economic constraints, directly impact the selection of a medium. Indeed, participatory approaches are highly dependent on the context in which the medium is inscribed (users, temporalities, scales, policies) [69]. A holistic estimation should, therefore, be considered in the selection of the medium, the evaluation framework offering only one concern of the issue.

Each medium offers benefits and drawbacks, mainly when digital technologies are involved [72]. Mediums' disparities are illustrated in this study in Figure 3. Some individuals from the targeted public could be more attracted to one medium than another because of personal or socio-demographic inclinations. Furthermore, each medium's assets and limits are generally complementary. For instance, an on-site participation asset brings experts into the discussion, where their relationship with the public is straightforward, without go-betweens. Their presence around the table consolidates the public's perception of the authorities' will to answer citizens' apprehensions, resulting in improved trust and commitment from the population. Experts in online participation are often more distant, and it can be laborious for participants to have an answer. One of the reasons is that an expert cannot be connected 24/7. The second is, considering a broader population reached by digital medium, every post cannot find a reply due to their large number. The improvement of trust and commitment described previously is, therefore, limited. A medium limit can be strengthened by another medium and vice versa. Based on this consideration, an appropriate combination of mediums (digital or traditional) should improve the overall participation approach and the targeted public reachability [61,69,73].

## 7. Conclusions

The gradual institutionalization of participatory approaches encourages the incorporation of local stakeholders in the decision-making arena. This involvement is challenging for local authorities that should, on the one hand, identify a targeted public for the scope of an interactive session, and on the other hand, engage these individuals through a specific medium. These challenges can quickly discourage the local officials that are lacking expertise and tools to make the right choice. This paper aims to bridge some of this gap by introducing an access evaluation framework articulated around the notion of access. Seven dimensions are presented from a participatory perspective: accessibility, availability,

adequacy, affordability, acceptability, awareness, and attractiveness. This new tool aspires to support the authorities during their selection of the best suited medium alternative to reach a targeted public. Moreover, this framework, if adopted after an interactive session, can stimulate a continuous improvement process. The organization of urban participatory approaches is certainly not limited to public access, therefore, the use of this tool should be included in a holistic evaluation, which considers other aspects, such as costs or risks.

Accelerated by the SARS-CoV2 pandemic, the digital transformation of participatory approaches introduces considerable stakes for the public. The promise of a genuine and broadened participation seems appealing; however, regarding a targeted public not proficient with digital technology, this potential could turn sour. Therefore, identifying the appropriate individuals supported by the local authorities is a crucial step in the participatory process, followed by the equally important selection of the right medium (enhanced by the access evaluation framework). A skillful practice of these two steps increases the likelihood of giving everyone from the targeted public equal access to the participatory processes. This could support the concerned citizens in expressing their opinion on urban issues, limiting bias and increasing the fairness of the decision, and ultimately contributing to the success of the participatory approach and the urban project.

## 8. URLs

CityPlanner: beta.cityplanneronline.com/site (last accessed on the 9 July 2020); NeoCity: neocityview.com (last accessed on the 9 July 2020); Open Government Partnership: open-govpartnership.org (last accessed on the 9 July 2020).

**Author Contributions:** Conceptualization, Thibaud Chassin and Adeline Cherqui; literature review and investigation, Thibaud Chassin and Adeline Cherqui; writing—original draft preparation, Thibaud Chassin; writing—review and editing, Thibaud Chassin, Adeline Cherqui, Jens Ingensand and Florent Joerin; supervision, Jens Ingensand and Florent Joerin. All authors have read and agreed to the published version of the manuscript.

**Funding:** This research received no external funding.

**Institutional Review Board Statement:** Not applicable.

**Informed Consent Statement:** Not applicable.

**Acknowledgments:** The icons used to build the figures originate from the Noun Project. Figure 1: "Cancel Download" and "Start Download" by Ranah Pixel Studio, *Place* by Hrbon, *Puzzle* by Jokokerto, *Problem* by Adrien Coquet, *Media* by Nithinan Tatah, *Communication* by Shashank Singh, *Puzzle* by Visual Language, *Puzzle* by Akshar Pathak, *Settings* by Danishicon, *Choice* by Dong Ik Seo and *Team Selection* by B Farias. Figure 2: *Report* by Becris, *People* by Benjamin Harlow, *Panoramic Landscape* and *Music Composer* and *Online Survey* by ProSymbols, *Mayor* by Laurent Patain, *Algorithm* by Eucalyp, *Writer* by Adrien Coquet and *City* by Mohkamil.

**Conflicts of Interest:** The authors declare no conflict of interest.

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
