# Peer review of "Impact of Digital and Non-Digital Urban Participatory Approaches on Public Access Conditions: An Evaluation Framework"

_ijgi, doi:10.3390/ijgi10080563_

Round 1

Reviewer 1 Report

I commend the authors for undertaking work in the important realm of participatory research. However, I cannot recommend this article for publication.

The article is difficult to read, the objectives and methods are unclear, and it is not grounded in the existing literature.

  1. Throughout, but in particular in the Introduction, there are many bold claims and sweeping generalizations that are not supported by references or examples. It's unprecedented to get to the 3rd page of a manuscript with only 3 references.
  2. We don't get to the objective of the article until pg. 5, but it's still unclear what the authors are trying to accomplish. The reporting of orginal results is very secondary and this largely appears to be a thought experient of how stakeholders might behave. There are so many assumptions and specifics that the potential to generalize findings is small.
  3. The access dimensions that the authors describe all appear in exisiting literature (which again, is not cited). The comparison of mediums (Table 3) is an interesting idea, but does not allow for nuance and local conditions (for example, assumes that everyone has access to a smart phone).
  4. Figure 3 - what are these values based on? Are these results or a thought experiment?
  5. Language should be simplified and made more clear throughout.

If the authors choose to revise, I would recommend a complete re-structuring that focuses on the case study (section 5.2) and uses those results to establish a framework. A comprehensive grounding in the participatory research and participatory mapping literature is essential. 

Author Response

Dear reviewer,

We appreciate your valuable comments provided on our work. We have revised our manuscript according to your comments. Please see the attachment to look at the changes and refinements made in the revised manuscript.

Please let us know if you have additional questions or concerns about the manuscript.
We will be happy to address them.

Sincerely,
The authors of the paper

Reviewer 2 Report

The article introduces a framework for evaluating public access and participation in urban planning projects. It then uses the framework to assess the involvement of the public based on an analysis of two participatory projects.

The proposed framework for evaluating public involvement is quite innovative but the details and essence of the work have not been clearly articulated in the article. The problem I find is that the article has been written by non-native English authors and so the manuscript must be edited regarding a) sentence construction – unclear and incomplete sentences, for example, use of appropriate words to describe events – several words are used inappropriately, e.g. 73: Two concomitant phenomena characterize public participation… and the flow of thought needs reorganization e.g., the discussion on online and offline mediums of participation should occur in the discussion of results of the study and not after that.

Author Response

(The authors gave the same response as above.)

Reviewer 3 Report

General comments

The paper presents a quite correct framework for evaluating different mediums of public participation and its application in two different case studies.

At present form I find that authors address the topic of public participation in a very neutral manner, as if there are no conflicting interests and power asymmetries in public participation processes. Besides, at some point it would interesting to distinguish between general public and affected or interested stakeholders (such as business associations or residents’ associations). Later on, in the second case study, this fact is mentioned, but I would recommend reviewing previous sections with these lens (see specific comments below).

Apart from that, I would also recommend to consider rewriting the title, because the concept of ‘public access’ it is not that clear until you read the article.

Specific comments:

Introduction

Please clarify the following sentences and concepts:

Line 45: “this article addresses this medium for structuring citizen participation.” Which medium? Do you mean ‘their medium’? The ones used in the two case studies?

Lines 46-47: “…this article highlights the fundamental issues of citizens’ access to these approaches.” What do you mean by citizens’ access? Access to information? / to the participatory process?... Besides what approaches are those that you referred? The medium used in both cases?

Later on, lines 185-188 explain better these ideas. I would suggest revising the previous sentences in order to make it clearer at the introduction as well.

2.2. Digital technologies supporting participatory approaches

Authors mentioned several experiences of collaborative mapping or platform that gathered citizen’s information, opinions… From a methodological point of view, it would also be interesting to mention other experiences that consider citizen participation during the whole process of the project/platform design such as:

ISPRS Int. J. Geo-Inf. 20187(2), 68; https://doi.org/10.3390/ijgi7020068

ISPRS Int. J. Geo-Inf. 20209(2), 130; https://doi.org/10.3390/ijgi9020130

In that sense, these works distinguish between collaborative and participative mapping methods.

  1. Access barriers in participatory approaches

Although I agree with most ideas presented in this paragraph, I miss some references to the fact that people participating sometimes have conflicting interest and definitively different power position (the asymmetries of power between stakeholders in participation processes is a classic in the literature). I would recommend mentioning at least the fact of conflict of interests between stakeholders. Maybe these ideas could be linked to sentences 175-179.

7 4. A framework for evaluating citizens’ access

Acceptability & Atracctivenes: From my point of view these terms do not take sufficient account of the interest of people in participating. From the perspective of the political ecology, this is an important issue and usually motivation is addressed as ‘interest’, and/or stake on the mater. People participate because they have a certain interest on the matter, and in most cases conflicting interest. As I mentioned before, the conflict of interest is an important issue in participatory processes that I think authors should explicitly mention.

Figure 3.: I would recommend placing it after or between its description, for example after line 434. On the other hand, the figure has 5 levels of evaluation and the table shows 3. Maybe you you should refer to section 4 contents here as you do with figure 5.

Figure 5.: Same recommendation: if it is possible place it after it has been mentioned (after line 531 in this case).

Line 588: to ‘antagonistic knowledge’ I would add ‘antagonistic interests’ and power asymmetries which from my point of view are key factors in the search for participation methods.

Line 621: I would also add that, in many times, authorities are also reluctant to lose power in decision-making processes

Author Response

(The authors gave the same response as above.)

Reviewer 4 Report

This is an interesting paper on the evaluation of public access in participatory urban planning processes. The theme is topical, approach well-founded and authors knowledgeable about relevant literatures. Yet another issue is that new systematic approaches to the evaluation of public participation are still needed, even though some existing approaches are already available.

Main comments:

You could raise in the introduction more clearly the issue that your evaluative framework will only cover one particular aspect of participation, which can and should be evaluated through more holistic evaluation frameworks, especially, when decisions are to be done about the worth of different participatory models, and which one of them to apply. In consequence, you might how the evaluation of access can help making better decisions on the method, since it only gives understanding of a very limited aspect of participation. (Perhaps the answer is that your evaluation framework is more oriented at planners who will continue designing and developing more functioning models; not so much for policy makers, who need to take into account many other aspects, such as the overall costs and risks of such mechanisms).

In the title of your paper you refer to digital transformation of participatory approaches. The criteria of your model, however, accessibility, for example, only refer to spatial aspects, which seems contradictory to the digital focus. Please consider either revising the criteria to cover digital aspects better, or (perhaps more likely, as you cannot probably change the criteria afterwards) consider framing the title/focus through other than digital focus.

As for the criteria, it seems a little artificial to keep on using criteria that only use a-letter (even this again can be well reasoned due to the reference to another model of similar sort). I found adequacy, for example, something that could be expressed more accurately by using some other words, such as time-wise etc. In case that you cannot again change the words, perhaps you could reflect this point/challenge/path dependency while describing the approach.

What I was missing here was the difference between different populations/segments. Attractiveness, for example, is constituted very differently to younger vs. older generations. However, you only provide one figure for all. Don’t you think it reasonable or at least informative to take into account these segment specific differences, while providing evaluation; in my view modern evaluation should be not only multi-variable but also multi-perspective.

As regards the application cases, for the latter you mention in the title that it a was an a posteriori evaluation; for the first case you could make it more clear in which stage the evaluation was carried out and how it was linked to the actual method design. Also, what were (potentially) the different evaluation of different groups.

So, overall, I think this paper adds an interesting approach to access evaluation. As I request you to contextualize to more holistic evaluation frameworks, perhaps equality of participation (or equal access to it) might be a relevant link word to such broader words (fairness, competence, quality, relevance, etc being some other big themes).

Some minor issues:

You claim in the first sentence that participation has been growing for 30 years. However, it is an movement in Western countries since 1960s

“district dread” is not fully clear concept

When you discuss “attractiveness”, you might perhaps start from the motivations and then continue to limitations/ entry costs

Figure 3 is interesting but brown colour/transect walks cannot be followed fully

Check the sentence: “The municipality representatives selected for this project an association of tran467

sect walks combined with an online survey”

Author Response

(The authors gave the same response as above.)
